# Knowledge, Attitudes, and Practices of Nurses toward Risk Factors and Prevention of Falls in Older Adult Patients in a Large-Sized Tertiary Care Setting

**DOI:** 10.3390/healthcare12040472

**Published:** 2024-02-14

**Authors:** Saad Mohammad Alsaad, Mshari Alabdulwahed, Nabeel Mohammed Rabea, Shabana Tharkar, Abdulaziz A. Alodhayani

**Affiliations:** 1Department of Family and Community Medicine, College of Medicine, King Saud University, Riyadh 11461, Saudi Arabia; abalodhyani@ksu.edu.sa; 2University Family Medicine Center, Department of Family and Community Medicine, College of Medicine, King Saud University Medical City, King Saud University, Riyadh 11461, Saudi Arabia; imesharya@gmail.com; 3Nursing Department, King Fahad Cardiac Center, King Saud University Medical City, King Saud University, Riyadh 19910, Saudi Arabia; nrabea@ksu.edu.sa; 4Prince Sattam bin Abdulaziz Research Chair for Epidemiology and Public Health, Department of Family and Community Medicine, College of Medicine, King Saud University, Riyadh 11461, Saudi Arabia; stharkar@ksu.edu.sa; 5Health Promotion and Health Education Research Chair, Department of Family and Community Medicine, College of Medicine, King Saud University, Riyadh 11461, Saudi Arabia

**Keywords:** knowledge, attitude, nurses, falls, aged, Saudi Arabia

## Abstract

The objective was to assess the knowledge, attitudes, and practices of nurses toward the prevention of falls in older hospitalized patients. A cross-sectional study employing a 54-item questionnaire was conducted on 370 nurses at a tertiary care referral center. The mean age of the study population was 36.3 ± 7.7 years, with the majority being females (282; 76.8%). Most of them had attended fall prevention training (335; 90.5%). More than 98% knew fall prevention policies and safety goals, according to their response to a fall and risk assessment, but were less aware of the risk factors of falls, such as recurrent falls (61%), depression (44%), and lower-extremity numbness (40.5%). Similarly, 99% had positive attitudes toward risk assessment, fall prevention intervention, and response to a fall. Around 55% thought they were responsible for patients’ falls, and 96% felt the need to undergo more training on fall prevention. Furthermore, 92% strictly followed fall prevention policies and 85.4% followed the color-coding system for high-risk patients. Despite the preventive measures in place, 33% encountered patient falls, and 82.2% experienced unwitnessed patient fall incidents in their units. Although the nurses had higher levels of knowledge about the policies, they lacked information on the risk factors. There is a significant scope that warrants great attention concerning the adherence to guidelines and the provision of fall prevention training programs, with a focus on the intrinsic causative factors of falls.

## 1. Introduction

A fall is defined as, “an event which results in a person coming to rest inadvertently on the ground or floor or other lower level”. Falls commonly occur in older adult populations during hospitalization. Falls are mostly accidental and unintentionally caused by oneself or others, but at times can be anticipated, especially during moments of agitation. The consequences and impact of falls during hospitalization exert a significant burden on the lives of older adults, which in turn affects various aspects of their lives. Their quality of life is severely affected, as patients are rendered heavily dependent on external assistance due to the physical impairments caused by falls and the prolonged duration of hospital stays, thus increasing costs. According to the Global Report on Falls Prevention in Older Age by the World Health Organization, inpatient care amounts to 50% of the costs of fall recovery, and long-term care costs amount to about 9.4–41% of the total costs involved, in addition to the huge burden of 40% of geriatric deaths resulting from injuries caused by falls [1]. Older adult patients contribute to a significant proportion of the population seeking nursing and special care. According to the world population projection of the United Nations, one in six people in the world will be over the age of 65 years by 2050, in comparison to one in eleven in 2019 [2]. The number of people over the age of 65 years is expected to double from 703 million in 2019 to 1.5 billion by the year 2050 [2]. In the Kingdom of Saudi Arabia, the older adults accounted for 4.2% of the total Saudi population in 2019, which is expected to rise by almost four times by the year 2050, accounting for 16.2% of the population [3,4]. The global prevalence of falls in individuals aged 65 years and over ranges from approximately 28% to 35%, increasing from 32% to 42% [1]. The prevalence of falls among the older adult Saudi population is 49.9% [5]. Between 5% and 10% of older adults who experience falls incur serious injuries, including fractures, head injuries, and serious lacerations [6]. Many studies have identified factors such as multiple morbidities, a decline in cognitive status and abilities, balance and gait alteration, visual impairments, and the use of prosthesis and/or orthosis as major contributors to biological risk factors [7,8]. Another cross-sectional study conducted among the older adult population over 60 years in the United Arab Emirates suggested polypharmacy, which means the use of five or more medications due to the presence of multiple comorbidities in the aging population as one of the major risk factors associated with increased fall risk [9].

Nurses play a pivotal role in preventive measures that have proven effective against the risk of patients’ falling. Their knowledge of the risk factors is crucial to enhancing the effectiveness of prevention strategies and mitigating the occurrence of falls. Evidence suggests that falls are influenced by nurses’ positive attitudes and knowledge toward fall prevention education [10,11]. In an observational study by Asiri, only 71% of the surveyed health centers in Saudi Arabia provided preventive measures to reduce fall risk [12]. Furthermore, the study reported that out of the twenty-four nurses surveyed, six displayed limited knowledge and understanding of fall prevention. It is essential to understand the level of knowledge and attitude of nurses toward fall prevention guidelines in their practice, which forms the rationale of the present study.

We hypothesize that the nursing staff registered in the university hospital have varying levels of knowledge regarding fall prevention in older adult patients, with a significant portion lacking comprehensive understanding. Hence, we aim to assess the knowledge of nursing staff registered in a university hospital on fall prevention in older adult patients, to provide insights to help reduce the occurrence of falls, thereby enhancing the quality of services in geriatric health care.

## 2. Materials and Methods

### 2.1. Study Description

An online cross-sectional survey was used to determine the knowledge, attitudes, and practices of nurses attending to older hospitalized patients at a leading tertiary care university hospital. The study was conducted for ten months between December 2020 and September 2021. The university hospital is one of a kind, with state-of-the-art infrastructure to cater to the needs of a diverse population. It is fully equipped with large diagnostic laboratories, surgical theatres, designated outpatient care, and an inpatient facility with more than a thousand bed capacity for hospitalization and inpatient care.

### 2.2. Study Sample

Convenient sampling was used to recruit the participants. The sample size was calculated using Raosoft^®^ (Sample Size Calculator, Raosoft Inc., Seattle, WA, USA) with a 5% margin of error, 95% confidence interval, a population size of 1832 nurses and a response distribution of 50%. A minimum of 318 participants were needed to be recruited for this study. However, the study recruited 370 subjects in the final sample. The inclusion criteria of study participants include the nurses employed at the medical and surgical wards while nurses from the outpatient clinics were excluded. 

### 2.3. Description of Study Questionnaire

The study adopted a valid and reliable questionnaire to elicit the participants’ data and responses regarding knowledge, attitudes, and practices related to fall prevention among the older hospitalized patients. The questionnaire was adapted from a previous study by Ganabathi and Umapathi that was conducted in the western part of Saudi Arabia [13]. The questionnaire consisted of four major sections that assessed (i) sociodemographic content, (ii) knowledge, (iii) attitude, and (iv) practices of the nurses.

Section I included questions on age, gender, years of experience, nationality, highest educational level, and fall prevention training. In section II, twenty items examined the knowledge levels of nurses regarding fall prevention (range = 0 to 20). Twenty items investigated the nurses’ attitudes toward fall prevention in section III, and the responses were recorded on a Likert scale as strongly agree, agree, disagree, and strongly disagree. Finally, section IV included fourteen items exploring the nurses’ practices related to fall prevention, describing the responses as never, rarely, sometimes, and always. To ensure the adaptability of the questionnaire in the current setting, a pilot test was conducted with a small group of nurses who were not part of the main study. Finally, Cronbach’s alpha was calculated and showed a high reliability score of 0.83.

### 2.4. Ethical Considerations

The study was approved by the Institutional Review Board (IRB) at the University Medical City (Ref. No. 21/0027/IRB) on 30 December 2020. In addition, complete anonymity was maintained to ensure the privacy and confidentiality of the study participants. Informed consent was included in the study questionnaire to ensure the voluntary participation of the nurses.

### 2.5. Data Collection Process

An online form of the questionnaire was generated using Google Forms. The head nurses of the medical and surgical wards were the first point of contact to be approached by the data collection team. The contact details of the nurses were obtained from the respective wards and the questionnaire link was sent requesting consent and participation in the survey. Reminder e-mails were sent to improve the response rate until the sample size was achieved. The data from Google Forms were observed for all kinds of missing data and necessary steps were taken to improve the quality of data in the Excel sheet.

### 2.6. Data Analysis

The data from the Excel sheet were exported to the Statistical Package of Social Sciences (SPSS) version 23.0 (IBM SPSS Statistics, Armonk, NY, USA). Descriptive statistics like frequencies, percentages, means, and standard deviations were used to elicit the responses of the items in the questionnaire.

## 3. Results

A total of 370 nurses were enrolled in the study. The sociodemographic characteristics of the study participants are shown in Table 1. The mean age of the study participants was 36 years, with a standard deviation of 7.7. Females (282, 76.2%) constituted the major portion of the study sample. Most of them had more than five years of work experience (286, 77.3%), and 265 (71.6%) had attained a bachelor’s level of education. The majority of the participants had attended a fall prevention training program (335, 90.5%).

The nurses’ knowledge about fall prevention is displayed in Table 2. More than 98.9% (*n* = 366) of the participants were very well aware of the fall prevention policies in the hospital, as a safety goal of the Joint Commission International Accreditation, response to accidental falls, identifying high-risk patients, and preventing a fall, as one of their most important duties. Likewise, the majority were aware of reporting a fall incident, the need for an interdisciplinary team to manage the patient in case of a fall incident, the likelihood of hip fractures in older adults, the increasing rate of hospitalization, and mortality associated with fall incidents. However, only 61%; (*n* = 226) [95% CI 56.04–65.95] were aware that past fall episodes are associated with increased fall recurrence. Furthermore, lower knowledge prevailed in associating depression (44%; *n* = 163) [95% CI 39.05–49.15] and numbness (40.5%; *n* = 150) [95% CI 35.62–45.61] with an increase in fall risk. Detailed statistics are explicitly shown in Table 2.

The nurses’ attitudes toward fall prevention are illustrated in Table 3. More than 99% showed a positive attitude toward risk assessment (*n* = 368), preventive education (*n* = 368), and fall prevention intervention (*n* = 367) and offered immediate help during a fall incident (*n* = 367). Although 86.8% (*n* = 321) felt that falls are unavoidable, 55% (*n* = 204) perceived that nurses are responsible for patients’ fall incidents and 77.8% (*n* = 288) showed a positive attitude toward prioritizing necessary intervention. Regardless of the 30% (*n* = 111) who felt that the consequence of falls was not severe, still, 96% (*n* = 355) of the participants desired more training in fall prevention.

The description of practices toward fall prevention in the hospital is depicted in Table 4. Good practices are taken by combining the responses “sometimes” and “always” and poor practices include “never” and “rarely”.

Fall prevention policies are strictly followed by 91.6% (*n* = 339) of the nurses, and 85.4% (*n* = 316) regularly follow the color-coding system for high-risk patients. Around 33% of the nurses have witnessed incidents of falls in their units. However, 82.2% (*n* = 304) stated having encountered unwitnessed patient fall experiences. The majority of the study participants have worked on preventing falls by either following policies or by educating patients (87.8%; *n* = 325), enhancing footwear (90.3%; *n* = 336), or accompanying the patient to the toilet (88.6%; *n* = 328). Inter-departmental help in terms of physiotherapy (81.1%; *n* = 300), occupational therapy (83.5%; *n* = 309), and social support (88.1%; *n* = 326) are sought by the majority.

## 4. Discussion

In this study, we investigated the knowledge, attitudes, and practices of nurses toward fall prevention activities for older patients. The findings of this study have important implications for improving patient safety during hospitalization by imparting educational and regular training programs in areas identified as knowledge-deficient. Although the study found overall higher knowledge levels, positive attitudes, and good practices toward fall prevention, considerable gaps do exist that must be addressed to meet higher standards of geriatric care.

Fatal and non-fatal injuries increase after a fall incident. Although 70% of falls result in minor and non-fatal injuries, deaths caused by fall incidents among senior people are still a major concern [13]. In countries like the United States, 9.9 million fall-related injuries occur each year, of which 32% are older adults with a higher mortality rate of 40% [14]. National studies from the Emergency Medicine Department indicated that India and China have reported 25% mortality rates and even higher [15,16]. Long-term mortality rates due to falls in older adults have increased considerably from 9.4 to 13.7 per 100,000 people in the last decade in China [17]. Furthermore, European nations have also reported an annual rise of 20% in fall-induced mortality rates in the older population [18]. Non-fatal and minor injuries due to falls constitute a much higher proportion, causing disability and levels of lower function. Hence, the prevention of falls during hospitalization and by healthcare workers, especially nurses, who are pioneers in lowering fall incidents, is given utmost importance.

### 4.1. Nurses’ Knowledge Regarding Fall Prevention

The present study was performed in a large organization that follows the safety goals set by the Joint Commission of International Accreditation (JCIA). In this study, we found that nurses were equipped with a good level of knowledge; however, scope exists for a better understanding of the results of certain items. Nurses had deep knowledge concerning fall prevention guidelines, policies, committees, risk assessments, and after-effects of a fall, but they lacked important information about the risk factors and causes related to the falls. Higher awareness about fall prevention guidelines may be attributed to the availability of fall prevention policies that are set as one of the safety goals through JCIA. Educating health workers in general and nurses in particular about patient fall prevention and identifying high-risk patients is a significant requirement for the Saudi Central Board for Accreditation of Healthcare Institutions (CBAHIs). On the contrary, knowledge about the causative factors of falls deserves special mention. Recurrent falling by itself is a risk factor for falls, the knowledge of which was only reported by 61% of the nurses. Similarly, sliding increases the risk of falling and was largely undermined by nurses. Research has demonstrated strong evidence linking falls with increased medication, polypharmacy, depression, comorbidities, and lower-extremity sensory deficits [19]. All of these risk factors were not well recognized by the nurses in this study, which represents some of the major findings. We recognize that the key point to the success of fall prevention programs is to improve the nurses’ understanding of the potential risk factors of patients’ falls. A review of the effect of nurse training on patients’ fall prevention demonstrated a significant reduction in fall rates and incidents, in addition to a positive change in patients’ behavior [20].

In comparison, similar findings were observed in international studies from the Middle Eastern region as well as other Western and Asian countries. A recent study from Egypt reported that nurses have less knowledge about the risk factors of falls [21]. In 2020, a Korean study explicitly demonstrated the differences in knowledge between fall prevention policies and the causes of falls [10]. These findings are consistent with our findings, which point toward improving strategies and knowledge dissemination in areas identified as deficient or limited, such as the causes and risk factors of falls. Educating nurses about the common associated factors of falls might significantly lower the risk of fall incidents in hospitals.

### 4.2. Nurses’ Attitude toward Fall Prevention

Our findings have demonstrated a positive attitude toward fall prevention. The nurses were generally concerned about their patients’ falls and felt that the responsibility of preventing a fall was shouldered by the nurses. Willingness to help patient mobility inside a room and prioritizing preventive education was valued to a large extent. However, they did not show a strong positive attitude toward interventions related to fall prevention. A significant proportion of the nurses showed a negative attitude toward the safety of the hospital environment and the severity of the physical injuries resulting from the falls. An expectation of a multidisciplinary approach in fall intervention might be one of the main reasons for this finding. In addition, they might feel overwhelmed with routine nursing duties and may consider patient safety and intervention as a shared responsibility. These results are consistent with other studies that have reported positive attitudes toward fall prevention but negative attitudes toward fall management or intervention [22,23].

Nurses’ attitudes toward the need for more extensive training programs in fall prevention is another highlight of this study. Although 85% of the nurses had the confidence to prevent a fall, they still desired more training. These findings are of clinical significance because frequent training in fall prevention has been directly correlated with a lower incidence of falls during hospitalization [19]. Despite the presence of training programs through JCIA and CBAHI, more education is needed to increase the knowledge and positive attitudes of nurses, particularly toward risk identification, fall prevention, and immediate management.

### 4.3. Nurses’ Practices toward Fall Prevention

On average, the present study showed a good level of practices adopted by nurses in patient safety and fall prevention. The level of knowledge and positive attitude may have a direct effect on good practices focusing on fall prevention. For large-sized organizations, there is still room to improve practices targeted at fall reductions. Although fall prevention policies have been followed strictly due to enforcement from the JCIA and safety committees, adherence to guidelines can be improved. For instance, the findings showed that 85% of the nurses followed the color-coding system to identify high-risk patients, and 87% provided education to their patients. These results are encouraging but can be further substantiated by increasing adherence. It is important to note that 122 (33%) of the nurses reported having experienced fall incidents in their units, and 82% claimed that the patient’s fall was not witnessed by staff. Yet, again, these findings are of utmost importance because, despite being equipped with optimal knowledge, positive attitudes, and good practices, falls are still being reported. This opens up an opportunity to improve adherence to fall prevention practices.

Our results are consistent with the findings reported by Ganabathi and Mariappan, who found similar practices in the western region of the Kingdom of Saudi Arabia [13]. However, inconsistency prevails from other studies that demonstrated poor practices.

## 5. Limitations

One of the limitations of the study is the low external validity and hence limited generalizability. First, since the data were collected from a single center, the study participants may not be representative of larger populations. Hence, a multi-centric study would add credibility to the existing results. Second, the questionnaire used was quite long, with 54 items, which may have increased fatigability, potentially causing inaccuracy in responses. Third, since the questionnaire was self-reported, the nurses may have had difficulty understanding and answering the questions. Fourth, the convenient sampling technique may introduce bias in the generalizability of findings. Finally, the social desirability bias may conceal the nurses’ responses to questions and their real opinions which may distort the results to a smaller extent.

## 6. Conclusions

The present study has widened our understanding of the existing literature on the knowledge, attitudes, and practices of nurses on fall prevention in older hospitalized patients. The study identified several gaps in the knowledge and practices of nurses about the risk factors and prevention of patient falls. These gaps can be addressed through targeted training programs that focus on the intrinsic causative factors of falls and by strengthening the curriculum and translating theory to practice. Policymakers can also play a pivotal role in ensuring that the guidelines are in place to improve the adherence of nurses to fall prevention policies. Additionally, regular audits and assessments can be conducted to identify areas that deserve improvement and to ensure best practice consistently. These efforts may substantially decrease the incidence of falls, thereby reducing the associated risks of prolonged hospitalizations, comorbidities, and mortality. In addition, addressing the deficient areas of positive attitude via education, motivation, and reinforcement may enhance efforts targeting reductions in fall incidents. Further studies involving larger sample sizes from multi-centric organizations may provide valuable data on organizing multifaceted and comprehensive fall prevention programs and strategies.

## Figures and Tables

**Table 1 healthcare-12-00472-t001:** Sociodemographic characteristics of the participants.

Variable	*n* = 370 *n* (%)
**Age in years (Mean ± SD)**	36.3 ±7.7
**Gender**	
Male	88 (23.8)
Female	282 (76.2)
**Years of Experience**	
1–5 years	84 (22.7)
6–10 years	142 (38.4)
11–15 years	66 (17.8)
More than 15 years	78 (21.1)
**Nationality**	
Saudi	91 (24.6)
Non-Saudi	279 (75.4)
**Highest Educational Level**	
Diploma	99 (26.8)
Bachelor	265 (71.6)
Master or PhD	6 (1.7)
**Did any of the training cover fall prevention**	
Yes	335 (90.5)
No	35 (9.5)

SD: Standard deviation, PhD: doctorate in philosophy.

**Table 2 healthcare-12-00472-t002:** Nurses’ knowledge regarding fall prevention.

Item	Yes *n* (%)	No *n* (%)
Recurrence rate is high among anyone who has already experienced a fall	226 (61.1)	144 (38.9)
There are fall prevention (FP) policies in this hospital	368 (99.5)	2 (0.5)
Is there a fall committee team in this hospital?	350 (94.6)	20 (5.4)
Are you aware that preventing falls is one of the safety goals of the Joint Commission International Accreditation?	365 (98.6)	5 (1.4)
Do you know what happens when a patient falls?	366 (98.9)	4 (1.1)
I always report the fall incident	357 (96.5)	13 (3.5)
The modified Morse Fall Risk assessment tool is used to assist you in identifying high-risk patients	365 (98.6)	5 (1.4)
Falls can affect the quality of life of patients	366 (98.9)	4 (1.1)
Falls can increase unnecessary acute care hospitalizations	355 (95.9)	15 (4.1)
Completion of a fall risk assessment to identify those patients at risk for falling	365 (98.6)	5 (1.4)
Nurses must collaborate with interdisciplinary team members and with patient/caregiver to be successful with FP	362 (97.8)	8 (2.2)
Fall prevention is an important aspect of my job	365 (98.6)	5 (1.4)
Falls increase an elderly person’s death rate	334 (90.3)	36 (9.7)
Hip fractures occur from falls	348 (94.1)	22 (5.9)
Sliding is not falling	158 (42.7)	212 (57.3)
The more medicine you take, the higher your fall risk	311 (84.1)	59 (15.9)
Depression is not related to falls	163 (44.1)	207 (55.9)
The more diseases you have, the higher your fall risk	335 (90.5)	35 (9.5)
Someone with a visual impairment has a higher risk for falls	366 (98.9)	4 (1.1)
Being numb in the limbs is not related to falls	150 (40.5)	220 (59.5)

**Table 3 healthcare-12-00472-t003:** Nurses’ attitudes regarding fall prevention (*n* = 370).

Item	Positive Attitude *n* (%)	Negative Attitude *n* (%)
I am concerned about patient falls	369 (99.7)	1 (0.3)
I think falls among patients is unavoidable	321 (86.8)	49 (13.2)
I think nurses are responsible for patients’ falls	204 (55.1)	166 (44.9)
I have concerns about interventions for FP	172 (46.5)	198 (53.5)
FP is a higher priority for intervention	288 (77.8)	82 (22.2)
Updates and education in the current trend of the FP is important in the FP program	359 (97)	11 (3.0)
I have to assess all the patients to determine risk factors for falls during admission and every shift	366 (98.9)	2 (1.1)
A patient’s fall risk level should be inspected when hospitalized	368 (99.5)	2 (0.5)
Fall preventive education is necessary	368 (99.5)	2 (0.5)
FP interventions should be performed actively	367 (99.2)	3 (0.8)
I will help immediately if someone asks for help when they move	367 (99.2)	3 (0.8)
The hospital environment is not safe for FP	367 (99.2)	3 (0.8)
Physical injury is not severe even if a fall happens	111 (30)	259 (70)
If injury occurs, falling has little significance	102 (27.6)	268 (72.4)
Fear of falling has a negative impact	124 (33.5)	246 (66.5)
I feel guilty if my patient falls	241 (65.1)	129 (34.9)
I have confidence in my ability to prevent falls	316 (85.4)	54 (14.6)
I need more training in fall prevention	355 (95.9)	15 (4.1)
Fall prevention should be a high priority	231 (62.4)	139 (37.6)
Falls occur because of patients	360 (97.3)	10 (2.7)

FP: Fall prevention; positive attitude = strongly agree and agree; negative attitude = strongly disagree and disagree.

**Table 4 healthcare-12-00472-t004:** Nurses’ practices regarding fall prevention.

Item	Good Practices *n* (%)	Poor Practices *n* (%)
Follow the color-coding system for high-risk fall patients.	316 (85.4)	54 (14.6)
Follow the FP policies strictly.	340 (91.9)	30 (8.1)
Participate in making FP policies.	280 (75.7)	90 (24.3)
Have you experienced any falls in your unit?	122 (33)	248 (67)
Offer/assist the patient to go to the toilet.	328 (88.6)	42 (11.4)
Encourage adequate hydration and nutrition.	336 (90.8)	34 (9.2)
Provide educational tools regarding FP to your patient.	325 (87.8)	45 (12.2)
When a patient fall is witnessed by staff and has no obvious injuries, how often is this reported?	326 (88.1)	44 (11.9)
When a patient reports a fall but is not witnessed by staff, how often is this reported?	304 (82.2)	66 (17.8)
Discuss ways to prevent falls.	339 (91.6)	31 (8.4)
Request physiotherapy to evaluate/treat strengthening, gait training, etc.	300 (81.1)	70 (18.9)
Encourage adequate footwear.	334 (90.3)	36 (9.7)
Seek occupational therapy to evaluate and provide instruction for the management of activity of daily living/instrumental activity of daily living.	309 (83.5)	61 (16.5)
Evaluate your patient for social support and resources such as funding for glasses/hearing aids.	326 (88.1)	44 (11.9)

## Data Availability

The data presented in this study are available on request from the corresponding author.

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
