# Peer review of "Knowledge, Attitudes, and Practices of Nurses toward Risk Factors and Prevention of Falls in Older Adult Patients in a Large-Sized Tertiary Care Setting"

_healthcare, 2024, doi:10.3390/healthcare12040472_

Round 1
Reviewer 1 Report
Comments and Suggestions for Authors
I would like to thank the authors for a very interesting article. This study evaluates the knowledge that nursing staff have about the prevention of falls in older adults with the aim of reducing the prevalence of falls and thereby improving the quality of genetic health care by professionals in the field. Of the health.
The abstract provides a complete and accurate overview of the purpose, methods, results and conclusions of the study. It is accessible to readers and meets standards for clarity and conciseness.
Although the topic of the article has great relevance in scientific literature, a series of weaknesses have been identified that must be modified for its acceptance:
Introduction:
- The introduction effectively establishes the motivation for the study and the research problem. A solid review of the literature is presented, but it is suggested to more explicitly highlight the knowledge gap that the study seeks to address.
- The first paragraph of the introduction section presents a definition of concepts, percentages and relevant information, but only a bibliographic reference repeated twice has been used. Other relevant citations must be introduced that support the information presented.
- Reference to polypharmacy as a risk factor could benefit from a brief explanation of how multiple medication use contributes to fall risk, providing more context for readers unfamiliar with the term.
- At the end of the same section the objective of the study is reflected, but it would be necessary to add its hypothesis(es).
Methods:
- The methods are described in a detailed and clear manner, which makes their replication easier. The inclusion of details about the participants, experimental design, and measured variables is appreciated. However, for a more complete understanding, it is suggested to expand the information and procedure.
Results:
The results are presented logically and supported by clear tables and graphs. However, you should review all tables and enter a table footer that reflects the abbreviations and their meaning such as SD, PhD or FP.
Discussion
The discussion could benefit from greater depth in the following aspects:
- The findings are mentioned as having "important implications for improving patient safety," but it would be useful to detail how these results could be translated into concrete measures to improve fall prevention practices in geriatric care.
-The discussion of mortality and injury rates from falls in different countries is relevant, but could be expanded by relating this data to the specific prevention practices adopted in those places. This would help to further contextualize the effectiveness of current practices in the sample studied.
- Discussion of gaps in nurses' knowledge is crucial, but could be more comprehensive by exploring possible reasons behind these deficiencies. Is it due to a lack of specific training, limitations in fall prevention policies, or other factors?
- The identification of risk factors, such as slipping and recurrent falls, is a key point. The discussion could be extended to explain how poor recognition of these specific factors could directly impact the effectiveness of falls prevention interventions.
- Although the need for more education is mentioned, it would be beneficial to discuss how existing training programs have affected fall prevention practices and whether they have had a measurable impact on reducing incidents.
Author Response
Reviewer #1
Comment: Introduction:
- The introduction effectively establishes the motivation for the study and the research problem. A solid review of the literature is presented, but it is suggested to more explicitly highlight the knowledge gap that the study seeks to address.
- The first paragraph of the introduction section presents a definition of concepts, percentages and relevant information, but only a bibliographic reference repeated twice has been used. Other relevant citations must be introduced that support the information presented.
- Reference to polypharmacy as a risk factor could benefit from a brief explanation of how multiple medication use contributes to fall risk, providing more context for readers unfamiliar with the term.
- At the end of the same section the objective of the study is reflected, but it would be necessary to add its hypothesis(es).
Response : Thank you for the positive comment. We have incorporated the necessary corrections as per the advice. [47-51], [65-68], [79-81].
Methods:
- The methods are described in a detailed and clear manner, which makes their replication easier. The inclusion of details about the participants, experimental design, and measured variables is appreciated. However, for a more complete understanding, it is suggested to expand the information and procedure.
Response : We appreciate the kind comment. Data collection section has been expanded.[121-127].
Results:
The results are presented logically and supported by clear tables and graphs. However, you should review all tables and enter a table footer that reflects the abbreviations and their meaning such as SD, PhD or FP.
Response: Footnote has been added now [140]
Discussion
The discussion could benefit from greater depth in the following aspects:
- The findings are mentioned as having "important implications for improving patient safety," but it would be useful to detail how these results could be translated into concrete measures to improve fall prevention practices in geriatric care.
-The discussion of mortality and injury rates from falls in different countries is relevant, but could be expanded by relating this data to the specific prevention practices adopted in those places. This would help to further contextualize the effectiveness of current practices in the sample studied.
- Discussion of gaps in nurses' knowledge is crucial, but could be more comprehensive by exploring possible reasons behind these deficiencies. Is it due to a lack of specific training, limitations in fall prevention policies, or other factors?
- The identification of risk factors, such as slipping and recurrent falls, is a key point. The discussion could be extended to explain how poor recognition of these specific factors could directly impact the effectiveness of falls prevention interventions.
- Although the need for more education is mentioned, it would be beneficial to discuss how existing training programs have affected fall prevention practices and whether they have had a measurable impact on reducing incidents.
Response : Thank you for the comment. We have incorporated all the aforementioned suggestions into the discussion section. [212-217]
Reviewer 2 Report
Comments and Suggestions for Authors
Fall is a preventative and useful quality indicator for hospitalized older adults.
Nurses are the primary gatekeepers and play a vital and prominent role in prevention of falls.
Very relevant study.
1. Introduction
“an event which results in a person coming to rest inadvertently on the
ground or floor or other lower level”
Comment: Falls can happen inadvertently and sometimes
accidentally or even intentionally by an agitated person mostly
a patient pushing or pulling others.
I suggest to modify the language to describe
all kinds of falls or add the word "mostly" before inadvertently.
5.Limitations
Acceptable and accurate identification of limitations. A multicentric study would add credibility to the results.
Author Response
Reviewer #2
Comments
Introduction
“an event which results in a person coming to rest inadvertently on the
ground or floor or other lower level”
Comment: Falls can happen inadvertently and sometimes
accidentally or even intentionally by an agitated person mostly
a patient pushing or pulling others.
I suggest to modify the language to describe
all kinds of falls or add the word "mostly" before inadvertently.
Response: Thank you for the comment. This line has been added in the introduction. [39-40]
- Limitations
Acceptable and accurate identification of limitations. A multicentric study would add credibility to the results.
Response: Thank you for the comment. This line has been added in the comment. [271-272]
Reviewer 3 Report
Comments and Suggestions for Authors
I would like to thank the authors for their work. This is an interesting paper, which aims to assess the knowledge of nursing staff registered in a university hospital on fall prevention in elderly patients.
Before the publication, I have some minor suggestions for the authors.
Abstract: I would suggest to add one or two sentences on the rational of this study.
Introduction: Some sentences are devoid of bibliographic citations, i.e. lines 39-43 and 66-68. I suggest to add at leat one citation for each sentence.
Methods: I would specify, among the data analysis section, whether and how the normality of the distribution of continuous variables was verified
Limitations: as an additional suggestion, please consider if your sample could be affected by the social desirability bias
Author Response
Reviewer #3
Comments
Abstract: I would suggest to add one or two sentences on the rational of this study.
Introduction: Some sentences are devoid of bibliographic citations, i.e. lines 39-43 and 66-68. I suggest to add at least one citation for each sentence.
Methods: I would specify, among the data analysis section, whether and how the normality of the distribution of continuous variables was verified
Limitations: as an additional suggestion, please consider if your sample could be affected by the social desirability bias
Response : Thank you for the comments. Since abstract has word limit, we are unable to add rationale in abstract but have added it in the last paragraph of introduction [81-84]. ND was not done since the study did not involve test of significance. We have addressed the social desirability bias in the limitations [276-278].
Reviewer 4 Report
Comments and Suggestions for Authors
Avoid the use of elderly. It is older adults
The introduction effectively sets the stage for the study but could benefit from a more detailed discussion on:
more insights into existing gaps in nurse training or knowledge regarding fall prevention to set the stage for the study's relevance.
ow this study's findings could lead to specific improvements in geriatric healthcare practices would strengthen the introduction.
Material and methods:
More information on the rationale behind convenient sampling and its potential impact on the study's generalizability would be useful.
Details on how the questionnaire was adapted and whether it was pilot-tested in the current setting would be valuable.
Mentioning any specific analytical strategies for handling missing or outlier data could improve transparency.
Results:
Overall, the results are well-presented
Discussion
The discussion effectively ties the study's results to the broader context of elderly care and fall prevention. However, it could benefit from a more in-depth analysis of how specific gaps in knowledge and practices among nurses can be addressed in future training and policy-making.
An expansion on how these findings could influence hospital policy changes or nurse training programs would enhance its utility.
Further elaboration on how these limitations might have affected the results and how they can be mitigated in future studies would strengthen the discussion.
The conclusion should be more concise while addressing the key aspects: Summary of Findings, Implications and Recommendations, Future Research Directions
Author Response
Reviewer #4
Comments and Suggestions for Authors
Avoid the use of elderly. It is older adults
Response: Thank you for the comments and suggestions. The word elderly has been rephrased with alternate synonyms. We have used older adults in the title based on your advice. Thank you.
Comment: The introduction effectively sets the stage for the study but could benefit from a more detailed discussion on:
more insights into existing gaps in nurse training or knowledge regarding fall prevention to set the stage for the study's relevance.
ow this study's findings could lead to specific improvements in geriatric healthcare practices would strengthen the introduction.
Response: This has been suitably added or rephrased accordingly in the introduction.
Comment:
Material and methods:
More information on the rationale behind convenient sampling and its potential impact on the study's generalizability would be useful.
Response: This is added in limitations.
Comment: Details on how the questionnaire was adapted and whether it was pilot-tested in the current setting would be valuable.
Mentioning any specific analytical strategies for handling missing or outlier data could improve transparency.
Response: This is already mentioned as Questionnaire was adapted from previous study with reference to no: 12, and to ensure the relevance and clarity of the questionnaire in the current setting, a pilot test was conducted with a small group of nurses who were not part of the main study.
A line on missing data has been added
Comment: Results: Overall, the results are well-presented
Response: Thank you for the comment
Comment: Discussion
The discussion effectively ties the study's results to the broader context of elderly care and fall prevention. However, it could benefit from a more in-depth analysis of how specific gaps in knowledge and practices among nurses can be addressed in future training and policy-making.
An expansion on how these findings could influence hospital policy changes or nurse training programs would enhance its utility.
Further elaboration on how these limitations might have affected the results and how they can be mitigated in future studies would strengthen the discussion
The conclusion should be more concise while addressing the key aspects: Summary of Findings, Implications and Recommendations, Future Research Directions
Response: Thank you for the comment. The discussion and conclusion have been suitably modified to improve credibility.
Thank you
Reviewer 5 Report
Comments and Suggestions for Authors
Dear authors, thank you very much for the opportunity to read your manuscript.
Since this is an observational study, the comments I am going to make are in accordance with the STROBE guidelines.
Abstract: You should include the objective of the research in the abstract.
Keywords: "Practices", "Patient fall" and "Elderly patients" are not Mesh terms. Keywords must be matched to descriptor terms for proper indexing of the manuscript.
Methods: In the participants section, the eligibility criteria should be described in greater detail, indicating the inclusion and exclusion criteria. Description of Study Questionnaire section, the dimensions into which the instrument is organized should be better defined. Reliability for each of the dimensions of the instrument, obtained in the aforementioned validation study in the context of Saudi Arabia. The functioning of the instrument should be explained (scores, range, likert response scale...). In the introduction section you must report the statistical results obtained with the cultural translation and valitation from the previous study in this context (items scores, standard deviations...).
Only descriptive tests (as shown in the results) are described in the statistical analysis section. I consider that the scientific interest of the research requires some inferential analysis using statistical tests appropriate to the normality of the distribution of the sample data. You should provide values of statistical significance, p-value, CI, effect size...
Results: Provide the "n" for the data in the results in the text. External validity is poor.
As the response items have not been explained in the methodology, the categories in Tables 3 and 4 (positive, negative; goog, poor) are not understood.
References: Include the link to the primary source or DOI.
Author Response
Reviewer #5
Comment
Dear authors, thank you very much for the opportunity to read your manuscript.
Since this is an observational study, the comments I am going to make are in accordance with the STROBE guidelines.
The Abstract: You should include the objective of the research in the abstract.
Response :Thank you for the comment. We have added the objective in the abstract [20-21].
Keywords: "Practices", "Patient fall" and "Elderly patients" are not Mesh terms. Keywords must be matched to descriptor terms for proper indexing of the manuscript.
Response: The keywords have been modified as per MESH [36].
Methods: In the participants section, the eligibility criteria should be described in greater detail, indicating the inclusion and exclusion criteria. Description of Study Questionnaire section, the dimensions into which the instrument is organized should be better defined. Reliability for each of the dimensions of the instrument, obtained in the aforementioned validation study in the context of Saudi Arabia. The functioning of the instrument should be explained (scores, range, likert response scale...). In the introduction section you must report the statistical results obtained with the cultural translation and valitation from the previous study in this context (items scores, standard deviations...).
Response: Thanks for the comment. We have now added every detail suggested by you as much as possible.
Only descriptive tests (as shown in the results) are described in the statistical analysis section. I consider that the scientific interest of the research requires some inferential analysis using statistical tests appropriate to the normality of the distribution of the sample data. You should provide values of statistical significance, p-value, CI, effect size...
Response: Thank you for the comment. We have dealt with only descriptive content in this paper.
Results: Provide the "n" for the data in the results in the text. External validity is poor.
As the response items have not been explained in the methodology, the categories in Tables 3 and 4 (positive, negative; goog, poor) are not understood.
Response: The results have now been explained in addition to explanation in methodology. We are grateful for your comments
References: Include the link to the primary source or DOI.
Response: Thank you for the comment. We have included the link to the primary source or DOI.
Thank you once again.
Round 2
Reviewer 1 Report
Comments and Suggestions for Authors
The authors have responded correctly to all the suggestions, so their publication is accepted.
Author Response
Thank you for your valuable comments and accepting our revised manuscript
Reviewer 4 Report
Comments and Suggestions for Authors
I would like to extend my congratulations on the significant improvements made to your manuscript. I think the revised version is now ready for publication.
Author Response

(The authors gave the same response as above.)

Reviewer 5 Report
Comments and Suggestions for Authors
Dear authors, thank you for the improvements in the manuscript. Some suggestions and comments from revision 1 that have not been corrected have yet to be addressed:
Keywords:
Keywords:keywords must correspond to MeSH terms for proper indexing of the manuscript.
Methods: Methods: In the participants section, the eligibility criteria should be described, indicating the inclusion and exclusion criteria. Description of Study Questionnaire section, the dimensions into which the instrument is organized should be better defined. Reliability for each of the dimensions of the instrument, obtained in the aforementioned validation study in the context of Saudi Arabia. The functioning of the instrument should be explained (scores, range...). In the introduction section you must report the statistical results obtained with the cultural translation and valitation from the previous study in this context (items scores, standard deviations...).
Results: Provide the "n" for the data in the results in the text. External validity is poor. As the response items have not been explained in the methodology, the categories in Tables 3 and 4 (positive, negative; goog, poor) are not understood.
Author Response
We thank the reviewer for his comments and we have carefully incorporated most of them and explained them point by point below.
Keywords:
Keywords: keywords must correspond to MeSH terms for proper indexing of the manuscript.
Response: Thank you for your comments. We have changed “Patients falls” to “Falls” and we have removed Practices from keywords. And replaced “Older patients” with “Aged”
Methods: Methods: In the participants section, the eligibility criteria should be described, indicating the inclusion and exclusion criteria. Description of Study Questionnaire section, the dimensions into which the instrument is organized should be better defined. Reliability for each of the dimensions of the instrument, obtained in the aforementioned validation study in the context of Saudi Arabia. The functioning of the instrument should be explained (scores, range...). In the introduction section you must report the statistical results obtained with the cultural translation and validation from the previous study in this context (items scores, standard deviations...).
Response: The inclusion and exclusion criteria has been added. The sections of the questionnaire have been further elaborated. Range is described for section I Total reliability is stated as Cronbach’s alpha.
Results: Provide the "n" for the data in the results in the text. External validity is poor. As the response items have not been explained in the methodology, the categories in Tables 3 and 4 (positive, negative; goog, poor) are not understood.
Response: n has been added in text now. External validity is explained in limitations. The categories in table 3 and 4 have now been described.
Thank you once again